# Fluoride Levels in Austrian Drinking Water Are Insufficient for Effective Caries Prevention

**DOI:** 10.3390/dj13100446

**Published:** 2025-09-29

**Authors:** Alice Blufstein, Elias Salzmann, Bledar Lilaj, Rinet Dauti, Oleh Andrukhov, Andrea Nell

**Affiliations:** 1Competence Center for Periodontal Research, University Clinic of Dentistry, Medical University of Vienna, 1090 Vienna, Austria; oleh.andrukhov@meduniwien.ac.at; 2Clinical Division of Periodontology, University Clinic of Dentistry, Medical University of Vienna, 1090 Vienna, Austria; elias.salzmann@meduniwien.ac.at; 3Clinical Division of Conservative Dentistry, University Clinic of Dentistry, Medical University of Vienna, 1090 Vienna, Austria; bledi.lilaj@gmail.com (B.L.); rinet.dauti@gmail.com (R.D.); nell.andrea@icloud.com (A.N.)

**Keywords:** fluorides, drinking water, dental caries, preventive dentistry, Austria

## Abstract

**Background/Objectives**: Fluorides play a well-established role in preventing dental caries, primarily by enhancing enamel resistance and inhibiting demineralization. Drinking water is among the most important sources of systemic fluoride intake. In 1993 and 2007, national analyses of Austrian drinking water revealed fluoride levels below 0.5 mg/L in almost all regions, which is insufficient for effective caries prevention. The present study aimed to re-examine the fluoride concentration in Austrian drinking water. **Methods**: Drinking water was collected in a total of 1985 Austrian municipalities. Fluoride concentration was measured by a fluoride-selective electrode. **Results**: The average fluoride concentration in the measured water samples ranged between 0.1 and 0.27 mg/L, depending on the region. The analysis revealed that 98% of the municipal drinking water samples contained fluoride at concentrations below 0.5 mg/L. In almost one quarter of Austrian municipalities, the fluoride levels amounted to less than 0.1 mg/L. The fluoride concentration in the drinking water of one Tyrolean municipality exceeded the recommended threshold. **Conclusions**: The results of the study reveal that the fluoride concentration in Austrian drinking water is generally too low to provide effective prevention against dental caries, affecting nearly all municipalities. Notably, the drinking water of one municipality reached potentially harmful fluoride levels. These findings could be used as a basis for targeted and individual fluoride supplementation, as well as for national or area-specific guidelines.

## 1. Introduction

Although dental caries is considered a largely preventable and controllable disease, its prevalence is still increasing globally and remains a major public health challenge [1]. In 2024, caries prevalence in 6- to 7-year-old children in Austria was 42%, which remains far from the World Health Organization (WHO) target of 80% caries-free children, corresponding to a maximum prevalence of 20% [2,3]. While there is no recent data for Austrian adults, the WHO reports a caries prevalence of 33.6% among the adult population in Europe in 2023. Apart from oral hygiene measures, the use of fluorides constitutes one of the most effective approaches to preventing dental caries [4]. Not only does it promote remineralization and prevent demineralization of teeth, but it also effectively inhibits the activity of cariogenic bacteria in dental plaque [5]. Fluorides can be applied either topically in the form of toothpastes, mouthwashes, varnishes, and gels, or systemically by the intake of tablets, lozenges, salts, and drinking water [6].

The fluoride concentration in drinking water depends on various factors, such as pH, temperature, the porosity and acidity of soil and rocks, total dissolved solids, and alkalinity [7]. Thus, high variations in fluoride levels can be observed: while in some parts of the world the naturally occurring fluoride concentration exceeds the threshold value of 1.5 mg/L as established by the WHO, other areas possess drinking water with fluoride levels below their recommended minimal concentration of 0.5 mg/L for preventing dental caries [8]. According to the United States Public Health Service, the ideal level to balance the benefits of fluorides on caries prevention while limiting harmful health effects is 0.7 mg/L [9]. In some countries, artificial fluoridation of water supplies is performed to achieve optimal fluoride levels. In Europe, however, community water fluoridation is rejected by most countries, including Austria [10]. In 1993, a nationwide analysis of the fluoride concentration in Austrian drinking water was conducted, revealing insufficient levels in almost all regions, which has also been observed in neighboring countries [11,12,13]. The last nationwide fluoride analysis of drinking water samples, which was conducted in 2007, not only reconfirmed the findings but also showed an even further decrease in the fluoride levels. At this time, a fluoride concentration higher than 0.5 mg/L has only been observed in the water samples of 30 municipalities [14].

It is important for dental professionals to consider local variations in the fluoride concentration in drinking water to provide adequate recommendations regarding topical/systemic fluoride application, and to avoid over-/underdosing. Therefore, the aim of the present study was to re-examine the fluoride concentration in Austrian drinking water nationwide.

## 2. Materials and Methods

Drinking water samples were collected in a total of 1985 municipalities in Austria. In each location, drinking water was collected within one day from three different sources, which were randomly chosen (private households, public restrooms, restaurants, shops, gas stations, etc.). After letting the water run for 30 s to avoid contamination, 500 mL of drinking water was collected in fresh 500 mL high-density polypropylene bottles (Flaschenland, Ransbach-Baumbach, Germany). Samples were stored at room temperature for a maximum of 14 days before analysis.

The fluoride concentration in the collected drinking water samples was analyzed in the Department of Analytical Chemistry, Faculty of Chemistry, University of Vienna, using a fluoride-ion-selective electrode (6.0502.150; Metrohm AG, Herisau, Switzerland), a Ag/AgCl reference electrode (3 M KCl, 6.0726.100, Metrohm AG, Herisau, Switzerland), and a 691 voltmeter (8.691.1001; Metrohm AG, Herisau, Switzerland). To calibrate the device and for further regression analysis, fluoride standard solutions were prepared with the following concentrations: 0.05, 0.1, 0.2, 0.5, 1, 5, and 10 mg/L. The detected voltage was plotted against the standard concentration for quantitative analysis via external calibration. From each sample, 50 mL of drinking water was mixed with 5 mL of the total ionic strength adjustment buffer, which regulates the ionic strength and pH of the sample. One liter of the TISAB II buffer contains 58 g NaCl, 3.0 g sodium citrate dihydrate, 57 mL acetic acid (100%), and pure water. Samples and standard solutions were stored for 24 h to achieve an equal temperature of 20 °C prior to analysis. Each sample was analyzed twice, and mean values were determined.

Data management, descriptive, and inferential statistics were conducted with Microsoft Excel 2016 (Microsoft Inc., Redmond, WA, USA) and GraphPad Prism 10 (GraphPad Software, Boston, MA, USA). Differences between fluoride level mean values in drinking water of the different regions were assessed by the ANOVA statistic with the Bonferroni post hoc test. The correlation between fluoride level and the caries prevalence was analyzed by the Spearman Rho method. The statistical analysis was performed with SPSS 27.0 (IBM, Armonk, NY, USA).

## 3. Results

### 3.1. Municipalities Included in the Study

Austria is structured into nine different regions, with Lower Austria having the largest area and Vienna having the smallest area. These regions are divided into a total of 2092 municipalities. As shown in Table 1, drinking water samples were collected in 1985 from 2092 Austrian municipalities.

### 3.2. Fluoride Concentration in Austrian Drinking Water

Figure 1 shows the mean fluoride concentration with 95% CI found in the drinking water of all nine Austrian regions. While Lower Austria had the highest mean fluoride concentration with 0.27 mg/L, the lowest values were observed in Vienna, Vorarlberg, Burgenland, and Salzburg, with average values below the detection limit of 0.1 mg/L. The values reached neither the optimal level of 0.7 mg/L established by the United States Public Health Service, nor the minimal concentration for carioprotective effects (0.5 mg/L) indicated by the WHO [7,8]. The regional variations in fluoride concentration in the drinking water of Austrian municipalities are demonstrated in Figure 2.

In total, only 38 of the 1985 included municipalities exceeded the minimal concentration of 0.5 mg/L, which represents 2% of all measured locations, as shown in Figure 3. Out of those municipalities, 16 locations showed fluoride levels above 0.7 mg/L, one of them (Karrösten, Tyrol) even exceeding the WHO threshold value with a concentration of 1.92 mg/L. Most of the municipalities, precisely 1505 (74%), exhibited drinking water with a fluoride concentration of 0.1–0.5 mg/L. Almost one quarter of the drinking water samples showed fluoride levels below 0.1 mg/L.

Figure 4 demonstrates the relationship between available caries prevalence data (children aged 6–7 years) and the fluoride concentration in the drinking water of nine Austrian regions [2]. Although a weak inverse relationship between these parameters can be observed, the analysis revealed no statistically significant correlation.

As presented in Table 2, the fluoride concentrations measured in 2007 and 2023 showed some relevant differences. The table contains all municipalities that showed fluoride levels above 0.5 mg/L in 2007, which was observed in 30 locations). In the recent analysis, 24 of these 30 municipalities had a lower fluoride concentration in their drinking water, 16 of them being below 0.5 mg/L. Out of the six municipalities with higher values in comparison to the assessment 15 years ago, one clearly exceeded the WHO guideline value of 1.5 mg/L.

Appendix A Table A1 demonstrates the analysis of the differences in fluoride concentrations in the drinking water between the nine Austrian regions, showing significant differences between numerous areas (*p* > 0.05).The exact fluoride levels in the drinking water of all 1985 municipalities are provided in Appendix A Table A2.

## 4. Discussion

Fluoride, the anion of the trace element fluorine, naturally occurs in water, air, and soil. Due to its chemical reactivity potential, fluorine is often bound to other chemical elements, e.g., metals of the first and second columns of the periodic table [15]. The amount of minerals and their solubility in drinking water primarily depend on the local, regional, and geological settings, as well as the hydro-geological conditions [16]. Austria has at least four different mountain/rock types, which contribute to the variety of fluoride concentrations found throughout the country [17,18]. While the fluoride concentration in the drinking water of the regions Vienna, Vorarlberg, Burgenland, and Salzburg was mostly below the detection limit, higher levels have been observed in areas like Lower Austria, Carinthia, Styria, and Tyrol. Similar tendencies have been observed in the analyses conducted in 1993 and 2007.

Overall, the drinking water of most municipalities analyzed in our study showed fluoride levels below 0.5 mg/L (98%), which is in accordance with the findings from the nationwide analyses in 1993 and 2007 [11,12,14]. Comparably low fluoride concentrations in drinking water have been observed in Germany. As reported by the German Federal Institute for Risk Assessment in 2025, 90% of German drinking water samples contain less than 0.3 mg/L of fluoride. A nationwide investigation from 2003 revealed similar results in Switzerland, showing fluoride levels below 0.3 mg/L in most cantons [13].

Low fluoride concentrations in drinking water could constitute an additional risk factor for developing caries, especially for individuals with a high susceptibility to dental cavities [19]. Caries is a widespread disease caused by a disbalance in the oral microbiome, a disturbance in the process of demineralization and remineralization, as well as a specific host response [20,21]. Since 2006, the Austrian National Public Health Institute (GÖG) has regularly assessed the dental status of Austrian 6- to 7-year-olds in some Austrian regions. The highest rates of children with caries experience were observed in Salzburg (2006: 69%) and Vienna (2006: 66%). Starting from 2016, the GÖG performed their evaluations in all Austrian regions, constituting the only source of data capturing caries prevalence nationwide. Comparing the analyses of 2016 and 2024, the proportion of Austrian 6- to 7-year-old children with caries experience slightly improved from an average of 45% to 42%. The lowest rates of children affected by dental caries were reported in Tyrol (2016: 30%; 2024: 28%), followed by Styria (2016: 34%; 2024: 32%). Interestingly, they are among the regions with higher fluoride levels in drinking water. Although the analysis revealed a weak tendency, no statistically significant correlation between caries prevalence and fluoride concentration in drinking water has been observed. This might be explained by the multitude of factors influencing caries prevalence, as shown by the GÖG, including sociodemographic factors. Although there was a major improvement in caries prevalence since 2006, the WHO goal of 80% caries-free children, or rather a maximum prevalence of 20%, is yet to be reached [2].

Apart from adequate oral hygiene and dietary measures, fluorides are considered one of the most important preventive actions against caries [22,23]. Their effectiveness is based on several different mechanisms: the presence of fluoride in dental plaque and saliva increases the remineralization of incipient enamel lesions and prevents demineralization. In an acidic environment, fluoride penetrates the enamel subsurface and forms crystals related to those of fluorapatite, which protects the enamel from dissolving. Furthermore, they interfere with the glycolysis of cariogenic bacteria and exhibit bactericidal effects in higher concentrations. Although the method of action is predominantly based on post-eruptive and topical effects, fluorides might increase the resistance of teeth against caries when ingested during tooth development. Apart from that, systemic fluoride intake also has topical effects by increasing the salivary fluoride concentration [24,25,26,27]. Accordingly, a substantial decrease in caries prevalence has been reported in areas with optimally fluoridated drinking water [28].

The beneficial effects of fluorides in proper amounts are undisputable; however, the total intake (topical and systemic) should be monitored to prevent overdosage [24]. As recommended by the WHO, the fluoride concentration in drinking water should not exceed the guideline value of 1.5 mg/L. Yet, the drinking water in the Tyrolean municipality of Karrösten contained 1.98 mg/L of fluorides, which might have potential adverse effects. At concentrations of 0.9–1.2 mg/L, mild dental fluorosis might occur, depending on the amount of other fluoride exposure sources [8]. These lesions are caused by long-term ingestion of higher fluoride levels during tooth mineralization and can range from small, white, almost unremarkable spots up to easily visible, large white areas of the enamel [26]. The only publication concerning dental fluorosis prevalence in Austria dates back to 1961, assessing 400 children aged 6 to 14 years old who were living in Mallnitz and other locations in the Ötz Valley with fluoride levels over 1.0 mg/L. In locations with 1.0 mg/L, the prevalence of fluorosis was 18.7%, reaching up to 59% in a municipality with 1.8 mg/L of fluoride in drinking water [29]. More recent data are available from the USA, where dental fluorosis prevalence has been repeatedly assessed for almost 40 years. While the prevalence has gradually increased from 22% in 1986/1987 to 87.3% in 2013/2014, it dropped to 68.2% in 2015/2016, possibly due to recommendations by the Department of Health and Human Services to lower water fluoridation in 2015 [30,31]. In contrast, a systematic review including international data revealed dental fluorosis prevalence of 40% at fluoride concentrations of 0.7 mg/L, although only 12% had aesthetic concerns [32]. However, all these studies suggest that higher fluoride levels are associated with a greater risk of dental fluorosis.

Higher levels of fluoride around 3–6 mg/L can lead to skeletal fluorosis, and at levels above 10 mg/L, a notably higher risk of bone fractures is described [8]. Prevalence data for skeletal fluorosis are not available for Europe and the USA, which might be due to the limited occurrence. Interestingly, a recent meta-analysis revealed an increased risk of skeletal fluorosis even at fluoride levels within the national guidelines [33]. In Austria, fluoride overdosage, especially intoxication, is highly unlikely, as long as the use of other fluoride sources is not exaggerated [12].

The present data could support dental professionals in individually advising their patients on the use of fluorides, depending on their place of residence. However, the present study is limited by possible seasonal and geological variations. Due to the dynamics of the fluoride concentrations throughout the years (as demonstrated in Table 2), measurements should be performed in regular intervals throughout the year to provide updated values. In addition, fluoride analyses of drinking water might be used as a basis for guidelines and recommendations by the Austrian health care system or the government for specific areas. These adjustments could reduce the need for restorative procedures in the dental office and, therefore, decrease the costs of the Austrian health care system. Furthermore, this may support the achievement of a paradigm shift from a reactive to an active, and thus preventive, strategy.

## 5. Conclusions

The present study revealed that the fluoride concentration in Austrian drinking water is below international recommendations for protective effects against dental caries in almost all municipalities, confirming the findings from 1993 and 2007. However, the fluoride levels in the drinking water of one Tyrolian municipality exceeded the WHO threshold value. These data should be taken into consideration for individual caries risk evaluation and may serve as a basis for targeted recommendations regarding fluoride supplementation.

## Figures and Tables

**Figure 1 dentistry-13-00446-f001:**
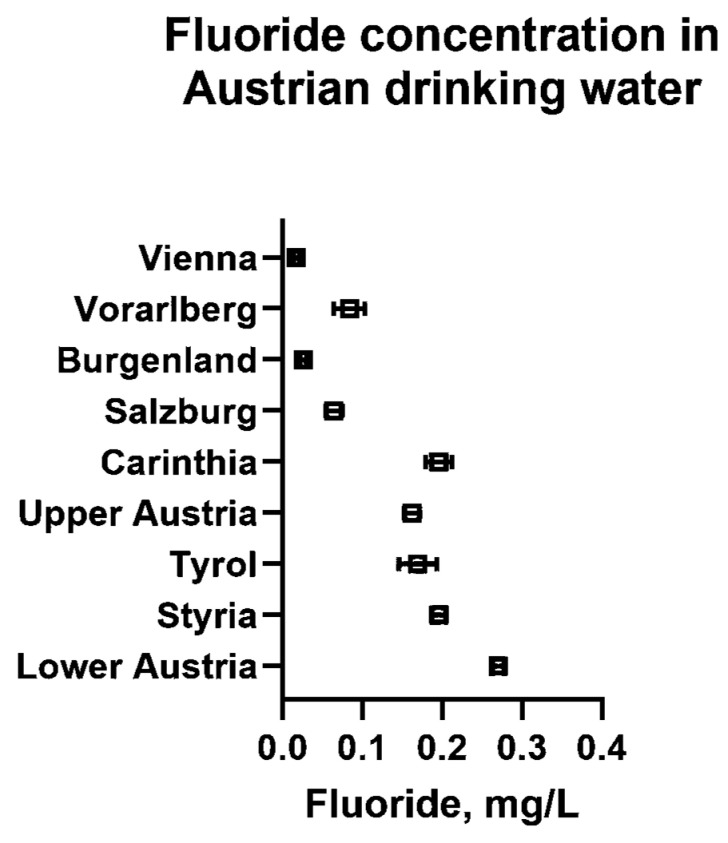
Mean fluoride concentration in Austrian drinking water per region. Data represent the mean values and 95% confidence intervals. Appendix A Table A1 demonstrates the analysis of the differences in fluoride concentrations in the drinking water between the nine Austrian regions.

**Figure 2 dentistry-13-00446-f002:**
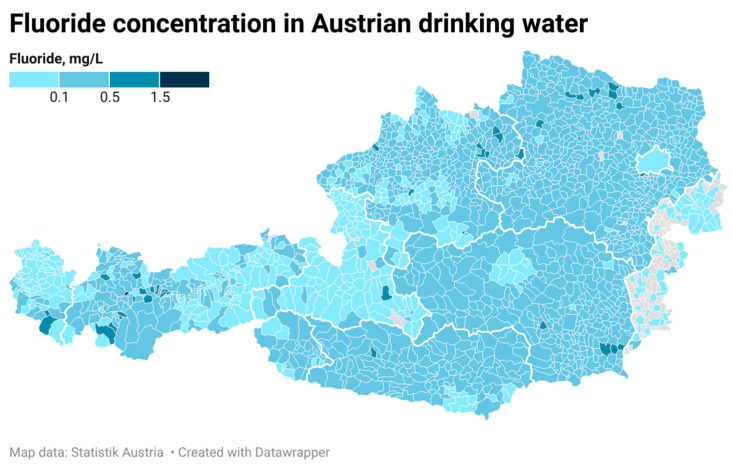
Regional variations in fluoride levels in the drinking water of Austrian municipalities.

**Figure 3 dentistry-13-00446-f003:**
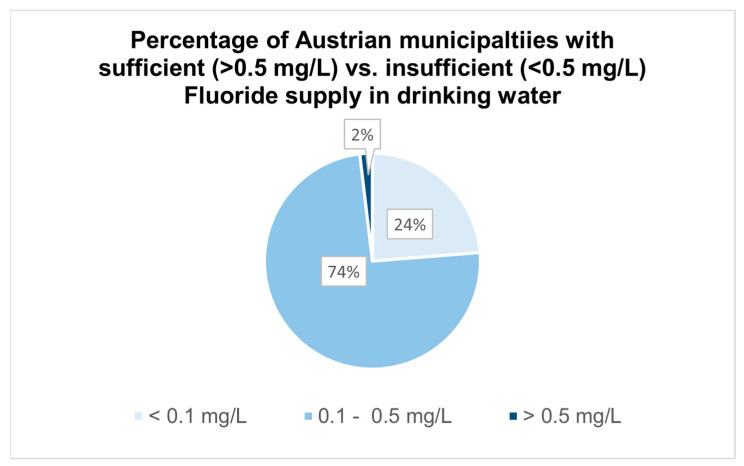
Percentage shares of Austrian municipalities with sufficient vs. insufficient fluoride concentration in drinking water.

**Figure 4 dentistry-13-00446-f004:**
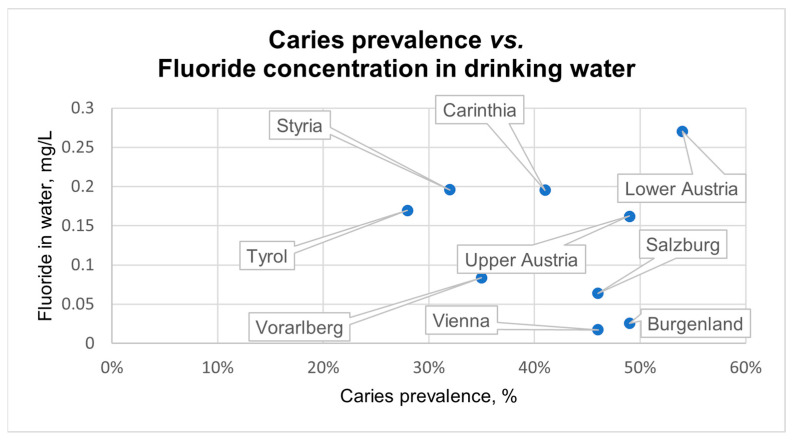
Relationship between caries prevalence of 6–7-year-old children and fluoride concentration in the drinking water of nine Austrian regions. A very weak, non-significant negative correlation between these parameters was observed (Spearman-Rho r = −0.126, *p* = 747).

**Table 1 dentistry-13-00446-t001:** Number of municipalities per region included in this study.

Province	Municipalities *
Lower Austria	573
Styria	286
Tyrol	277
Upper Austria	438
Carinthia	132
Salzburg	119
Burgenland	61
Vorarlberg	96
Vienna	3
**TOTAL**	**1985**

* included in the study (1985 out of 2092 municipalities).

**Table 2 dentistry-13-00446-t002:** Changes in fluoride concentrations in the drinking water of municipalities with levels > 0.5 mg/L in 2007.

	2007	2023	↑/↓		2007	2023	↑/↓
**Vorarlberg**				**Lower Austria**			
Innerbraz	0.65	0.33	↓	Payerbach	0.57	0.29	↓
St. Anton	0.51	0.35	↓	St. Georgen am Ybbsfelde	1.14	0.32	↓
St. Gallenkirch	0.64	0.51	↓	**Salzburg**			
Schruns	0.7	0.54	↓	Bad Gastein	2.35	0.17	↓
**Tyrol**				**Carinthia**			
Biberwier	0.88	1.26	↑	Mallnitz	0.53	0.20	↓
Elmen	0.53	0.99	↑	Oberdrauburg	0.59	0.30	↓
Faggen	0.53	0.34	↓	**Upper Austria**			
Haiming	1.37	<0.01	↓	Bad Zell	2.09	0.14	↓
Inzing	0.66	1.09	↑	Hargelsberg	0.92	0.11	↓
Karrösten	1.58	1.92	↑	Mehrnbach	0.95	0.16	↓
Kaunerberg	1.3	0.93	↓	Münzbach	1.24	0.39	↓
Oberhofen	0.5	0.61	↑	Ottenschlag im Mühlkreis	0.57	0.12	↓
Rietz	0.92	1.00	↑	Ried in der Riedmark	0.51	0.5	↓
Silz	1.71	0.85	↓	Schönau im Mühlkreis	0.63	0.59	↓
St. Leonhard	1.08	0.21	↓	Windhaag bei Perg	1.17	0.42	↓
Tösens	0.6	0.5	↓	Windischgarsten	0.69	0.57	↓

↑ Fluoride concentration in 2023 is higher than in 2007; ↓ Fluoride concentration in 2023 is lower than in 2007.

## Data Availability

The original contributions presented in this study are included in the article. Further inquiries can be directed to the corresponding author.

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
