# Peer review of "Fluoride Levels in Austrian Drinking Water Are Insufficient for Effective Caries Prevention"

_dentistry, 2025, doi:10.3390/dj13100446_

Round 1
Reviewer 1 Report
Comments and Suggestions for Authors
Congratulations to the authors on conducting this valuable research. Gaining insight into water fluoride concentration levels at the national level is truly important and interesting topic.
However I have some comments that should be considered.
1. Introduction
The introduction is clear and concise, and it nicely frames the research problem. It would be helpful if you could add another study that reports national-level caries prevalence in the general population, not only in children.
2. Materials and methods
How did you determine the sample size? Please specify.
3. Results
The results are well described and supported by graphs and tables.
4. Discussion
The discussion is well written. It would be helpful to include additional references and compare them with your results. Please also specify the limitations of your study.
Author Response
Comment 1: The introduction is clear and concise, and it nicely frames the research problem. It would be helpful if you could add another study that reports national-level caries prevalence in the general population, not only in children.
Response 1: Thank you for your valuable suggestion. Unfortunately, there is currently no recent data available concerning caries prevalence in the general population of Austria. However, we added the caries prevalence among adults in Europe to the introduction section.
Comment 2: How did you determine the sample size? Please specify.
Response 2: The sample size was not specifically calculated, as we aimed to include as many Austrian municipalities as possible. Thus, water samples have been collected in 1985 out of 2092 Austrian municipalities. The remaining 107 have not been included due to difficulties accessing the locations.
Comment 3: The results are well described and supported by graphs and tables.
Response 3: We appreciate your positive feedback.
Comment 4: The discussion is well written. It would be helpful to include additional references and compare them with your results. Please also specify the limitations of your study.
Response 4: Additional references have been added to the discussion section of the manuscript and compared with our findings. Furthermore, the authors thank you for pointing out the missing study limitations. They have been included into the revised version of the manuscript.
Reviewer 2 Report
Comments and Suggestions for Authors
IMHO there is no need to add the last long list of localities with their Fluoride content. It can be in an annex, or deposited in a webpage with a link in the paper.
Author Response
Comment 1: IMHO there is no need to add the last long list of localities with their Fluoride content. It can be in an annex, or deposited in a webpage with a link in the paper.
Response 1: Thank you for your valuable comment. We agree that the list is too long to include into the main text. Therefore, we submitted the table as an Appendix.
Reviewer 3 Report
Comments and Suggestions for Authors
Dear authors
Methodology
- Sampling information: Although municipalities were extensively represented, the paper does not specify if sampling was conducted in various seasons or at different water sources within each municipality. Seasonal or geological variations in fluoride concentrations could impact the outcomes.
- Storage and stability: The water samples were kept for as long as 14 days prior to analysis. While fluoride is typically stable, providing additional details (such as proof that storage did not affect the concentration) would enhance confidence in the results.
- Statistical analysis: The study relies solely on descriptive statistics (using Excel). There is no mention of inferential analyses (for instance, comparing means across different regions or providing confidence intervals). This restricts the strength of the conclusions drawn.
Contextual and Comparative Analysis
- Missing policy connection: The research underscores inadequate fluoride levels yet fails to assess Austria’s preventive measures (such as the use of fluoride toothpaste, salt fluoridation, and school-based fluoride initiatives). In the absence of this evaluation, the relationship to caries prevalence is overly simplified.
- Data on caries prevalence: The article makes a brief comparison between fluoride levels and caries prevalence (for instance, Salzburg versus Tyrol), but this is merely anecdotal and lacks statistical analysis. The authors could enhance their argument by linking regional fluoride concentrations to dental health outcomes.
Discussion Limitations
- Mechanistic repetition: The mechanisms of fluoride (such as enhancing enamel resistance, providing antibacterial properties, and facilitating fluorapatite formation) are thoroughly explained but mainly reiterate well-established information, contributing little new understanding.
- International comparison: Although the WHO thresholds are mentioned, drawing comparisons with neighboring EU countries (like Germany, Switzerland, and Hungary) would add valuable context.
- Risk of over-fluoridation: Although only one municipality surpassed the 1.5 mg/l mark, a more evidence-based discussion regarding the risks of fluorosis and skeletal impacts could be beneficial, focusing on prevalence rather than purely theoretical concerns.
Presentation & Data Transparency
- Tables and figures: Utilizing figures such as maps and distribution curves could more effectively demonstrate regional variations in fluoride compared to relying solely on tables.
All the best
Author Response
Comment 1: Sampling information: Although municipalities were extensively represented, the paper does not specify if sampling was conducted in various seasons or at different water sources within each municipality. Seasonal or geological variations in fluoride concentrations could impact the outcomes.
Response 1: Thank you for pointing out the missing information. The water samples were collected from three different sources per location. The collection in each municipality took place within one day. We added this information into the Material & Methods section of our manuscript. The fact that seasonal and geological variations did not receive attention in our study has been added as a limitation of our study.
Comment 2: Storage and stability: The water samples were kept for as long as 14 days prior to analysis. While fluoride is typically stable, providing additional details (such as proof that storage did not affect the concentration) would enhance confidence in the results.
Response 2: Thank you for raising these justifiable concerns. Our analysis was conducted in the Institute of Analytical Chemistry, University of Vienna. The fluoride standard solutions used in the laboratory are stored up to 6 months without exhibiting any changes in the fluoride concentration. Thus, we are positive that a storage of 14 days prior to analysis does not affect the measured concentration.
Comment 3: Statistical analysis: The study relies solely on descriptive statistics (using Excel). There is no mention of inferential analyses (for instance, comparing means across different regions or providing confidence intervals). This restricts the strength of the conclusions drawn.
Response 3: Thank you for pointing this out. The revised version of the manuscript contains additional statistical analyses. Firstly, a confidence interval was added to the mean values of the fluoride concentrations. Secondly, the correlation coefficient between available caries prevalence data (6-7 year olds) and fluoride concentration in drinking water of different Austrian regions has been calculated and added to the revised manuscript. Finally, a statistical analysis of the differences between the fluoride concentration mean values in the different areas is provided as supplementary material.
Comment 4: Missing policy connection: The research underscores inadequate fluoride levels yet fails to assess Austria’s preventive measures (such as the use of fluoride toothpaste, salt fluoridation, and school-based fluoride initiatives). In the absence of this evaluation, the relationship to caries prevalence is overly simplified.
Response 4: We are thankful you this valuable comment and agree that the evaluation of preventive measures in Austria might be of high interest. Certainly, fluoridated toothpaste and salt are available in Austria, however, not exclusively. The only way to provide data about preventive measures would be an evaluation in all municipalities, which is, unfortunately, hardly feasible. We further agree that the absence of these data simplifies the relationship to caries prevalence. However, we believe that the primary aim, a nation-wide analysis of fluoride concentration in drinking water, has been met with the present study.
Comment 5: Data on caries prevalence: The article makes a brief comparison between fluoride levels and caries prevalence (for instance, Salzburg versus Tyrol), but this is merely anecdotal and lacks statistical analysis. The authors could enhance their argument by linking regional fluoride concentrations to dental health outcomes.
Response 5: Thank you for your suggestion. As mentioned before, we calculated the correlation coefficient between the available caries prevalence data and fluoride concentrations Austrian regions and added the corresponding Figure to the Results section.
Comment 6: Mechanistic repetition: The mechanisms of fluoride (such as enhancing enamel resistance, providing antibacterial properties, and facilitating fluorapatite formation) are thoroughly explained but mainly reiterate well-established information, contributing little new understanding.
Response 6: We agree that the discussion contains well-established information about fluorides. However, we believe that our study might be of interest to individuals in non-dental fields and thus, providing thorough explanations might benefit the general reader.
Comment 7: International comparison: Although the WHO thresholds are mentioned, drawing comparisons with neighboring EU countries (like Germany, Switzerland, and Hungary) would add valuable context.
Response 7: Thank you for this valuable suggestion. We added information about neighbouring EU countries in the discussion section.
Comment 8: Risk of over-fluoridation: Although only one municipality surpassed the 1.5 mg/l mark, a more evidence-based discussion regarding the risks of fluorosis and skeletal impacts could be beneficial, focusing on prevalence rather than purely theoretical concerns.
Response 8: We are thankful for this helpful recommendation. The discussion was extended by Austrian and international prevalence data of dental fluorosis. Concerning the prevalence of skeletal fluorosis, there is no data available for Europe and the USA. This information was also included in the discussion section.
Comment 9: Tables and figures: Utilizing figures such as maps and distribution curves could more effectively demonstrate regional variations in fluoride compared to relying solely on tables.
Response 9: We agree that a map is beneficial for demonstrating regional variations. Therefore, a choropleth map of fluoride concentrations in drinking water of Austrian municipalities has been added in the revised version of the manuscript. In addition, the other figures have been revised.